# Radiographic interpretation using high-resolution Cbct to diagnose degenerative temporomandibular joint disease

Jonas Bianchi[1,2,3]*, João Roberto Gonçalves[3], Antônio Carlos de Oliveira Ruellas[2,4], Júlia Vieira Pastana Bianchi[1], Lawrence M. Ashman[5], Marilia Yatabe[1], Erika Benavides[6], Fabiana Naomi Soki[6], Lucia Helena Soares Cevidanes[1]

1 Department of Orthodontics and Pediatric Dentistry, School of Dentistry, University of Michigan, Ann Arbor, Michigan, United States of America, 2 Department of Orthodontics, University of the Pacific, Arthur A. Dugoni School of Dentistry, San Francisco, California, United States of America, 3 Department of Pediatric Dentistry, São Paulo State University (Unesp), School of Dentist, Araraquara, São Paulo, Brazil, 4 Department of Orthodontics, School of Dentistry, Federal University of Rio de Janeiro, Rio de Janeiro, Rio de Janeiro, Brazil, 5 Oral & Maxillofacial Surgery, Hospital Dentistry, University of Michigan, Ann Arbor, Michigan, United States of America, 6 Department of Periodontics and Oral Medicine, School of Dentistry, University of Michigan, Ann Arbor, Michigan, United States of America

* jbianchi@pacific.edu

**Data Availability Statement:** All relevant data are within the manuscript and its Supporting information files.

## Abstract

The objective of this study was to use high-resolution cone-beam computed images (hr-CBCT) to diagnose degenerative joint disease in asymptomatic and symptomatic subjects using the Diagnostic Criteria for Temporomandibular Disorders DC/TMD imaging criteria. This observational study comprised of 92 subjects age-sex matched and divided into two groups: clinical degenerative joint disease (c-DJD, n = 46) and asymptomatic control group (n = 46). Clinical assessment of the DJD and high-resolution CBCT images (isotropic voxel size of 0.08mm) of the temporomandibular joints were performed for each participant. An American Board of Oral and Maxillofacial Radiology certified radiologist and a maxillofacial radiologist used the DC/TMD imaging criteria to evaluate the radiographic findings, followed by a consensus of the radiographic evaluation. The two radiologists presented a high agreement (Cohen's Kappa ranging from 0.80 to 0.87) for all radiographic findings (osteophyte, erosion, cysts, flattening, and sclerosis). Five patients from the c-DJD group did not present radiographic findings, being then classified as arthralgia. In the asymptomatic control group, 82.6% of the patients presented radiographic findings determinant of DJD and were then classified as osteoarthrosis or overdiagnosis. In conclusion, our results showed a high number of radiographic findings in the asymptomatic control group, and for this reason, we suggest that there is a need for additional imaging criteria to classify DJD properly in hr-CBCT images.

## Introduction

Medical and dental imaging research on patient-specific diagnostics is a growing area, and the use of different imaging modalities has been reported, such as MRI [1], conventional X-rays

**Funding:** National Institute of Health (NIH), National Institute of Dental and Craniofacial Research (NIDCR) grant: R01DE024450.

**Competing interests:** The authors have declared that no competing interests exist.

[2], Computed Tomography, and Cone-Beam Computed Tomography (CBCT) [3–6]. Additionally, the development of new technologies and the improvement of existing imaging equipment facilitate the extraction of more precise and meaningful diagnostic information. Consequently, there is a need to manage and interpret these novel data to develop new therapies and diagnostic approaches [7–9].

In dentistry, CBCT is the exam of choice to evaluate many bone-related diseases [10–12]. The temporomandibular disorders (TMD) [13] is a broad term used to describe signs and symptoms that affect muscles and joints of the temporomandibular area. The recommended "Diagnostic Criteria for TMD (DC/TMD) Axis I protocol" include diagnostic criteria for differentiating the most common pain-related TMD [14]. In the temporomandibular joint (TMJ), the degenerative joint disease (DJD) is a degenerative disorder involving the joint characterized by deterioration of articular tissue with concomitant skeletal changes in the condyle, articular fossa, and eminence. Different terminology is associated with DJD: 'Osteoarthritis' is used to describe for any clinical and radiographic signs and symptoms associated with pain; 'Osteoarthrosis' is used when no clinical signs and symptoms are present, but has radiographic findings, and the term 'Arthralgia' is used when clinical signs and symptoms are present, but no radiographic findings [13, 14].

In the DC/TMD guidelines, the diagnosis for TMJ disc displacement with reduction and DJD should include history in the last 30 days of any TMJ noise present with jaw movement or function, or the patient reports any noise during the exam; however, the sensitivity and specificity for this diagnosis are only 0.55 and 0.61, respectively [14]. The confirmatory and the reference standard exam for the diagnosis is the Computed Tomography (CT) criteria which should contain one of the following radiographic signs: subchondral cyst(s), erosion(s), generalized sclerosis, or osteophyte(s). Flattening and sclerosis only are considered indeterminant factors for OA. However, the study that describes the radiological/imaging criteria published by Ahmad et al. [6] used Computed Tomography (CT) of the TMJ with a slice thickness of 1mm, and two diplomates of the American Board of Oral and Maxillofacial Radiology assessed the images showing the reliability of 0.71 to evaluate the osseous changes. However, as the spatial resolution of those images was relatively large (1 mm thickness), the influence of the voxel size in the diagnostic performance to assess TMJ bony changes in CBCT images has been widely discussed in the literature. Overall conclusion is that there is a need for more studies exploring the diagnostic quality and reliability of CBCT images [4].

In this study, we used a relatively new CBCT machine [15] using a high-resolution imaging protocol to obtain high-resolution cone-beam computed images (hr-CBCT) of the TMJs (0.08 mm of isotropic voxel-size). The aim was to use high-resolution cone-beam computed images (hr-CBCT) to diagnose degenerative joint disease in asymptomatic and symptomatic subjects using the Diagnostic Criteria for Temporomandibular Disorders DC/TMD imaging criteria. We also hypothesized that there are no differences between the two radiologists for the diagnosis of the degenerative joint disease in the sample.

## Material and methods

We followed the Strengthening the Reporting of Observational Studies in Epidemiology (STROBE) [16] guidelines for reporting observational studies.

### Study design, setting, participants, and ethics approval

This study was reviewed and approved by the Institutional Review Board of the University of Michigan (HUM00105204 and HUM00113199). A written consent form was obtained for each participant. This observational study was composed of human subjects recruited at the

University of Michigan (Ann Arbor–MI, USA) from January 2016 to July 2019. The sample comprised of prospectively recruited 46 clinically asymptomatic control patients (Control group) and 46 patients with a clinical diagnosis of degenerative joint disease (c-DJD group), resulting in 92 participants and 184 mandibular condyles.

### Sample size calculation

The sample size calculation for comparing groups was done using the software G-power [17] with α of 0.05, power (1- β) of 0.80, and Cohen's effect size given by the median and standard deviation of each group from a pilot sample. We performed the calculation for the imaging findings of osteophytes, erosion, and cysts. The size (n) necessary per group was 32, 44, and 58, respectively, and in this study, we had a final sample size of 92 per group.

### Clinical diagnosis, exclusion, and inclusion criteria

All participants were between 21–70 years of age, with no history of systemic diseases, jaw joint trauma, surgery or recent jaw joint injections, pregnancy, or congenital bone or cartilage disease. They were clinically evaluated by the same temporomandibular joint specialist, using the clinical signs and symptoms of the Diagnostic Criteria for Temporomandibular Disorders (DC/TMD). To be clinically diagnosed with degenerative joint disease (c-DJD group), they must present TMJ noise during movement or function in the last 30 days, and crepitus detected during mandibular excursive movements. In addition, they must have reported TMJ pain within ten years. On the other hand, for the control group, the subjects did not present any clinical signs and symptoms of DJD.

### Cone-beam computed tomography acquisition

Each participant had a high-resolution cone-beam computed tomography (CBCT) exam of each TMJ acquired using the 3D Accuitomo 170 (J. Morita MFG. CORP. Tokyo, Japan) scanner at the University of Michigan. The TMJ acquisition protocol used a field of view 40x40 mm; 90 kVp, 5 mAs, scanning time of 30.8 s, and a isotropic voxel size of 0.08 mm x 0.08 mm x 0.08 mm. The images were exported in DICOM (.dcm) format using the i- Dixel software (J. Morita MFG. CORP Tokyo, Japan) and were de-identified for further radiological evaluation. The cone-beam computed tomography (CBCT) exams for both groups were acquired respecting the ALARA and ALADA principles [17, 18]. All participants agreed and signed the informed consent term for participating in this study. In this study, the radiation dose to the patients was kept as low as possible by limiting the field of view (FOV) of the CBCT scan to 40 x 40 mm, which is enough to cover the TMJ area. The patients also wore a lead apron with a thyroid collar.

### Imaging diagnostic criteria and radiographic findings agreement

A multi-planar and blinded evaluation of the CBCT scans was performed by an American Board of Oral and Maxillofacial Radiology certified radiologist and a maxillofacial radiologist from the Department of Periodontics & Oral Medicine at the University of Michigan–School of Dentistry (Ann Arbor, USA). The multi-planar cross-sectional images were assessed in sagittal and coronal planes to score the mandibular condyle using the following five categories (as defined and adapted from the DC/TMD) and four scales (0 to 3): A) Flattening: 0 = Not visualized, 1 = Mild, 2 = Moderate and 3 = Severe; B) Osteophytes: 0 = Not visualized, 1 = Mild/Small, 2 = Moderate/Medium, 3 = Severe/Large; C) Sclerosis: 0 = Not visualized, 1 = Mild/Localized, 2 = Moderate/Generalized, 3 = Severe/Generalized; D) Erosion: 0 = Not visualized,

1 = Mild/Localized, 2 = Moderate, 3 = Severe, and E) Cysts: 0 = Not visualized, 1 = one to two cysts, 2 = three to four cysts, 3 = five or more cysts. Each CBCT scan was evaluated separately by each radiologist, followed by a consensus radiographic evaluation. The consensus data was used for interpretation and radiological classification, and if the patient presents any score for erosion, osteophyte and/or cysts, he was classified as having a radiographic diagnosis of degenerative joint disease.

## Statistical analysis

Our data showed nom-normal distribution assessed by the Shapiro-Wilk test. For this reason, the Mann–Whitney U test was used to compare the radiographic findings between the two groups (Fig 1C). To test observer concordance, an agreement matrix was calculated, and to test the inter-rater reliability, the Cohen's kappa coefficient (κ) was conducted. A description of the main findings was evaluated and presented in plots (Fig 1A and 1B) to visualize the

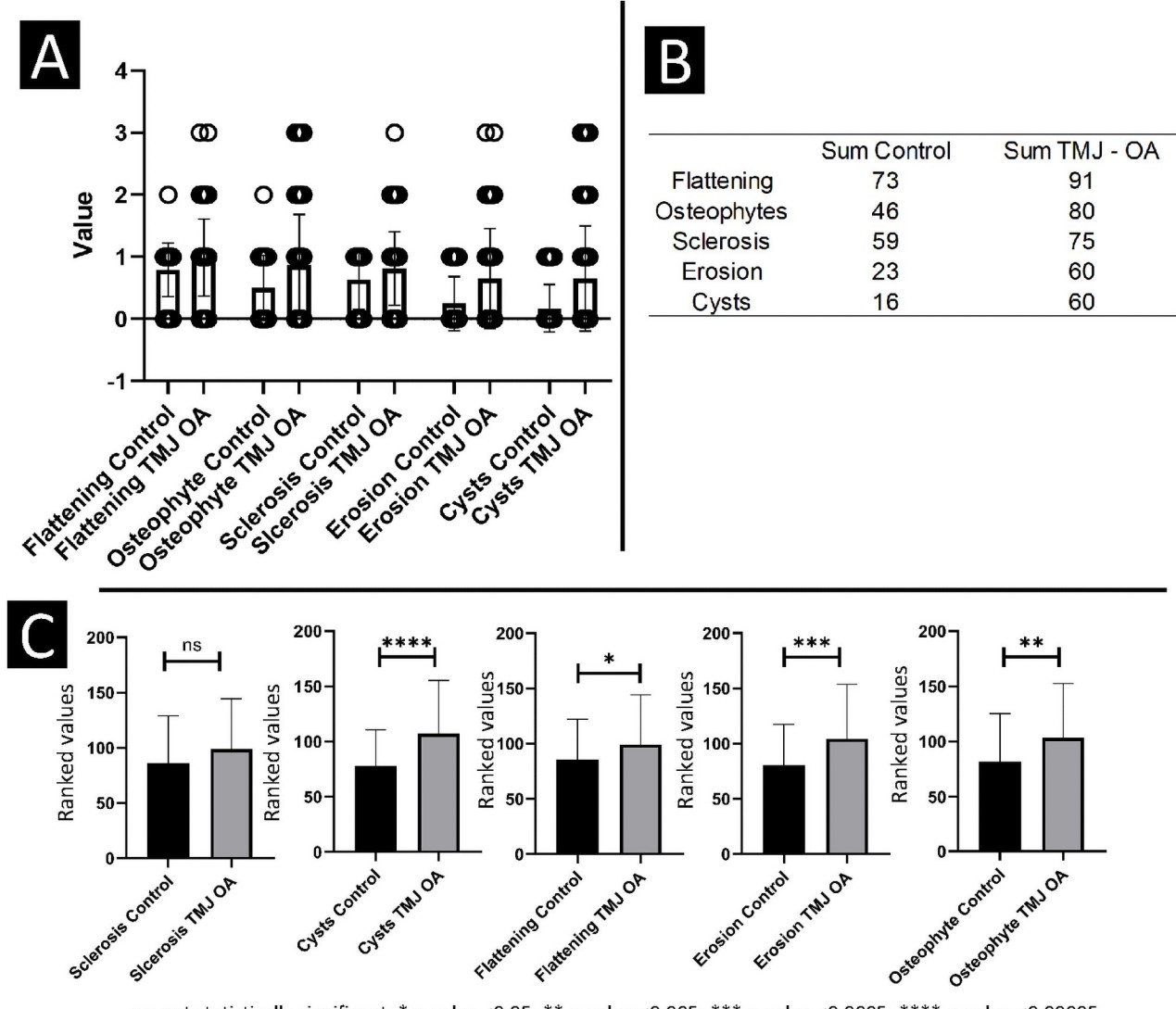

ns: not statistically significant; * p-value <0.05; ** p-value <0.005; *** p-value <0.0005; **** p-value <0.00005.

**Fig 1. A, B**—Descriptive statistics for the radiologists' consensus. **C**- Mann Whitney U test for comparison between the groups.

distribution of the radiographic findings between the groups and imaging findings (Fig 1A) and the sum of each radiographic finding (Fig 1B).

## Results

The sample was composed of sex-age matched subjects, resulting in 2 groups: c-DJD group and Control group. The c-DJD group (46 patients and n = 92 condyles) consisted of 39 females and 7 males, average age: 38.50 (S.D: 13.26), and the Control Group (46 patients and n = 92 condyles) has 39 females and 7 males, with a mean age of 38 years (S.D: 12.34). Table 1 shows the agreement matrix between the two observers for the radiographic findings of both groups together. It can be noted that for flattening, osteophytes, sclerosis, erosion, and cysts scores. The only disagreement was one above or one below score, and that the overall agreement was excellent. Table 2 shows the Cohen's Kappa statistic values for the measurements between the observers: the lowest κ value being 0.80 for sclerosis, and the highest κ being 0.87 for erosion.

The agreement between the radiologists' interpretation of the findings in each group is summarized in Fig 1A and 1B. The sum count of the classification used in this study included 0 to 3 scoring. All radiographic finding values were higher in the c-DJD group compared to the control group. As expected, flattening showed a high incidence in both groups, followed by osteophytes, sclerosis, erosion, and cysts. Thus, those findings showed that radiographic changes were observed in both groups, having a higher magnitude in the c-DJD group. Inter-group comparison shows that 'sclerosis' was the only radiographic finding that was statistically similar between both groups (p>0.05). The following radiographic findings showed that c-DJD group had significantly worse scores: flattening (p-value $\leq$ 0.05), osteophytes (p-value $\leq$ 0.005), erosion (p-value $\leq$ 0.0005), and cysts (p-value $\leq$ 0.00005) as illustrated in Fig 1C.

Fig 2 compares the percentage of condyles with no radiographic findings versus condyles with radiographic findings. Fig 3 summarizes this study's sample radiographic interpretation.

Fig 3 summarizes our main findings for patient classification. For c-DJD group, approximately 11% of the population did not show imaging signs of DJD. From 92 enrolled subjects, 46 presented clinical symptoms of DJD associated with TMJ pain, and 46 participants did not present any clinical sign and/or history of TMJ problems. In the radiographic interpretation, 41 patients showed signs of DJD whereas five did not. For these five patients, we suggested a diagnosis of Arthralgia and the other 41 patients received a diagnosis of DJD–Osteoarthritis. For the control group in the radiographic interpretation, 38 subjects showed signs of DJD, and only 8 did not, so based on the DC/TMD these patients should receive a diagnosis of Osteoarthrosis. Fig 4 shows the cross-sectional hr-CBCT and the radiographic findings: cysts (Fig 4A), localized erosions (Fig 4B), and osteophytes (Fig 4C) in both groups.

## Discussion

In this study, we proposed to evaluate the radiological interpretation of cone-beam computed tomography images with a voxel size of 0.08mmx0.08mmx0.08mm using the DC/TMD imaging criteria for DJD diagnosis. We showed that the two radiologists presented a good agreement between them (κ = 0.80). However, over 82% of the asymptomatic control patients showed a radiographic finding (Fig 2), which is a higher incidence than the current literature [6, 19–21], suggesting either a high occurrence of bone remodeling that may be over-diagnosed as osteoarthrosis or true incidence of osteoarthrosis in our study sample. It is possible that as the DC/TMD imaging criteria for DJD were validated for CT images with lower voxel size, there may be a lack of adequate imaging criteria to properly classify DJD using high-resolution CBCT images.

**Table 1. Agreement matrix of the radiologists (observer 1 and 2) for each radiographic finding visualized in a multi-planar HR-CBCT evaluation for both groups together.** The agreement is shown in diagonal. The numbers represent the total number of mandibular condyles in their specific category.

| Flattening | | Observer 2 | | | | Total |
|---|---|---|---|---|---|---|
| | | Not visualized | Mild | Moderate | Severe | |
| **Observer 1** | Not visualized | 30 | 4 | 0 | 0 | 34 |
| | Mild | 7 | 124 | 2 | 0 | 133 |
| | Moderate | 0 | 3 | 12 | 0 | 15 |
| | Severe | 0 | 0 | 0 | 2 | 2 |
| Total | | 37 | 131 | 14 | 2 | 184 |
| **Osteophytes** | | Not visualized | Mild / Small | Moderate / Medium | Severe / Large | Total |
| | Not visualized | 71 | 14 | 0 | 0 | 85 |
| | Mild / Small | 5 | 77 | 0 | 0 | 82 |
| | Moderate / Medium | 0 | 3 | 7 | 0 | 10 |
| | Severe / Large | 0 | 0 | 2 | 5 | 7 |
| Total | | 76 | 94 | 9 | 5 | 184 |
| **Sclerosis** | | Not visualized | Mild / Localized | Moderate / Generalized | Severe / Generalized | Total |
| | Not visualized | 49 | 14 | 0 | 0 | 63 |
| | Mild / Localized | 4 | 110 | 0 | 0 | 114 |
| | Moderate / Generalized | 0 | 1 | 5 | 0 | 6 |
| | Severe / Generalized | 0 | 0 | 0 | 1 | 1 |
| Total | | 53 | 125 | 5 | 1 | 184 |
| **Erosion** | | Not visualized | Mild | Moderate | Severe | Total |
| | Not visualized | 114 | 4 | 0 | 0 | 118 |
| | Mild / Localized | 5 | 45 | 1 | 0 | 51 |
| | Moderate | 0 | 3 | 9 | 1 | 13 |
| | Severe | 0 | 0 | 1 | 1 | 2 |
| Total | | 119 | 52 | 11 | 2 | 184 |
| **Cysts** | | Not visualized | 1–2 cysts | 3–4 cysts | 5 or more cysts | Total |
| | Not visualized | 120 | 6 | 0 | 0 | 126 |
| | 1–2 cysts | 7 | 38 | 4 | 0 | 49 |
| | 3–4 cysts | 0 | 1 | 6 | 1 | 8 |
| | 5 or more cysts | 0 | 0 | 0 | 1 | 1 |
| Total | | 127 | 45 | 10 | 2 | 184 |

**Table 2. Cohen's Kappa statistic to test inter-rater reliability between the radiologists.**

| Observer 1 x Observer 2 | | | 95% CI | | | |
|---|---|---|---|---|---|---|
| | Weighted Kappa | SE | Lower | Upper | Observed Agreements | Agreements Expected by Chance |
| Flattening | 0.83 | 0.05 | 0.71 | 0.90 | 91.30% | 55.81% |
| Osteophytes | 0.82 | 0.04 | 0.69 | 0.86 | 86.96% | 42.22% |
| Sclerosis | 0.80 | 0.05 | 0.69 | 0.88 | 89.67% | 52.04% |
| Erosion | 0.87 | 0.04 | 0.76 | 0.91 | 91.85% | 49.74% |
| Cysts | 0.81 | 0.05 | 0.68 | 0.87 | 89.67% | 54.02% |

CI: Confidence interval; SE: Standard error of Kappa.

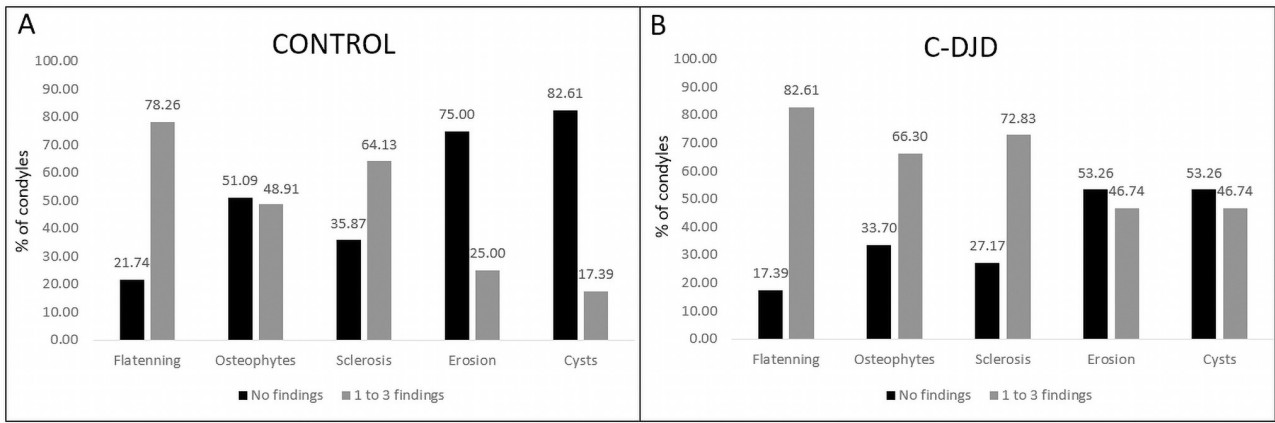

**Fig 2. Percentage of condyles with no radiographic findings (score 0, not visualized) compared to condyles with radiographic findings (scores 1–3 of severity grouped).**

The validation of CT images for diagnosis of DJD was proposed in 2009 to confirm the clinical diagnosis based on the patient's history and signs/symptoms because the clinical diagnosis only had a sensitivity of 0.55 and specificity of 0.61 [14]. The determinant imaging criteria for DJD proposed by Ahmad et al. used CT images with a slice thickness of 1mm and the following categories: subchondral cyst, and/or erosion, and/or osteophyte, and/or generalized sclerosis[6]. The challenge was the osseous defects had to be close to 1mm in order to be detected, making this diagnosis more reliable for late to chronic DJD stages. The greatest challenge now is to detect the disease before severe bone changes occur in the mandibular condyles, and

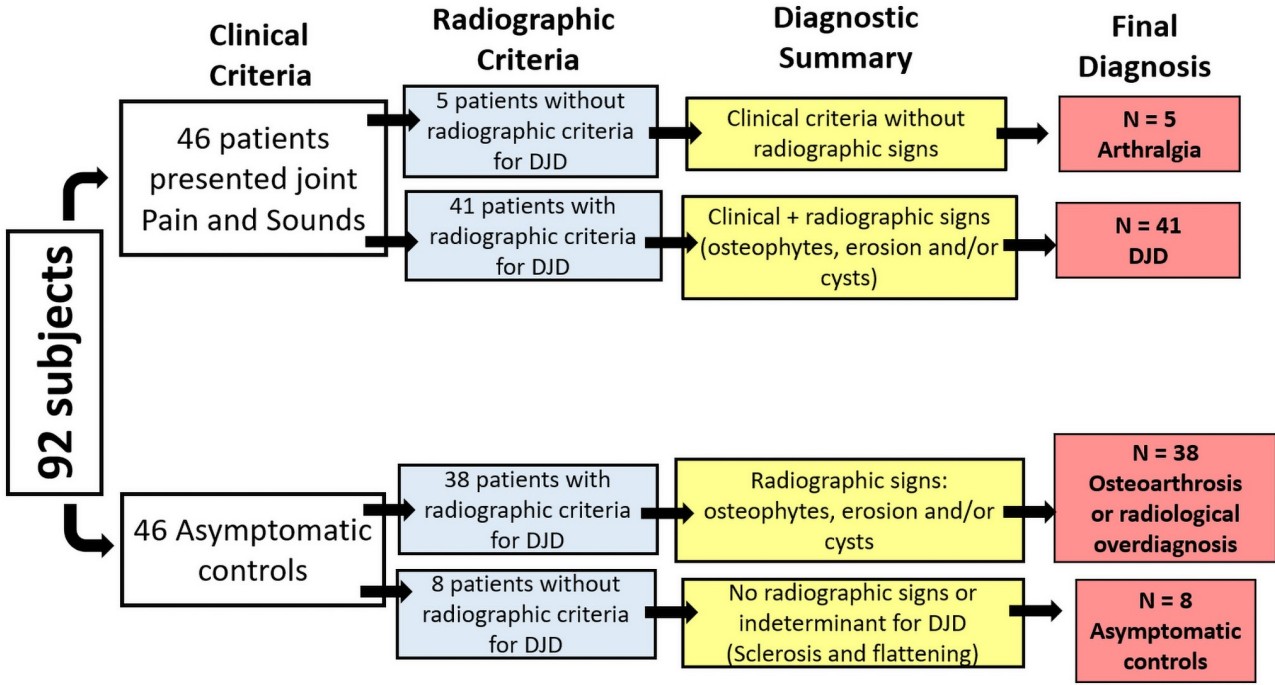

**Fig 3. Summary of patient's diagnosis after clinical and radiological assessment.**

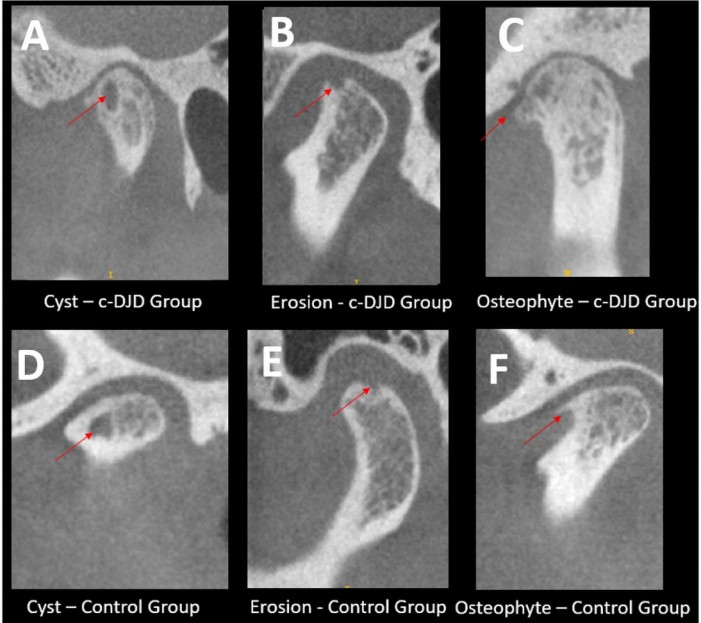

**Fig 4. Radiographic findings determinants for DJD.** Each image shows the sagittal slices of the TMJs and hr-CBCT with isotropic voxel size of 0.08 mm with the arrows are pointing to the finding. A-C: Represents the DJD group and D–F the asymptomatic control group.

studies indicate the key role subchondral bone plays in osteoarthritis progression [20–23]. In cross-sectional evaluation of the CT images, authors found a κ = 0.71 for reliability and 86% of the agreement for TMJ OA diagnosis[6]. In present study, our results are based on each radiographic finding that characterizes the DJD, and has showed an inter-rater agreement of 86% for osteophytes, 89% for cysts and sclerosis, and 91% for erosion and flattening (Table 2). The Cohen's kappa values were also high between 0.80 to 0.87 showing the radiologists were calibrated between them, and they could identify and correctly classify the imaging findings based on the DC/TMD guidelines.

Although CT imaging has contributed to the diagnosis of TMDs [14], in dentistry, most studies have used CBCT images to assess the bone changes, and the accuracy of CBCT for the detection of osseous defects remains under discussion [3, 20, 21, 24–26]. A systematic review reported the reliability of studies that used CBCT images to detect the osseous changes in TMJs and concluded that voxel size is a parameter that affects the pooled sensitivity (PSS) and pooled specificity (PSC), i.e., for isotropic voxel sizes of 0.2 mm or less the PSS is 0.73 and PSC is 0.68 and the studies that used voxel size of 0.4 to 0.5 the values are 0.83 and 1.00 respectively [4]. Another study has pointed out that voxel size is not important for the improvement of diagnosis; however, an image downsizing technique was used to increase the voxel size in that study and does not represent the real patient CBCT protocol acquisition [2]. A study from Lukat et al. [27] assessed the effects of voxel size to detect osseous changes in temporomandibular joint (TMJ), and they found no statistically significant difference between the voxel sizes in detection of TMJ osteoarthritic changes. However, a limitation was that the authors acquired the images with a isotropic voxel size of 0.076 mm and computationally downsized the image to a voxel size of 0.300 mm, which does not represent a true CBCT acquisition with 0.300 mm of voxel size.

In the present study, we have shown that using CBCT images with 0.08mm isotropic voxel resolution resulted in a good agreement in the diagnostic interpretation between the two observers. Furthermore, we found a high number of osseous changes in asymptomatic control group (38 out of 46 patients). These findings correspond to approximately 82% of the control group that presented radiographic signs of DJD. However, the literature has shown a smaller percentage of asymptomatic patients. Cevidanes et al. [28] found approximately 15% of changes in control patients using a isotropic voxel size of 0.5 mm. Krisjane et al. [29] assessed osteoarthritis findings in asymptomatic patients and found the most common signs were articular surface flattening and subcortical sclerosis; however, it was found only in 43% of class ll patients without clinical signs and symptoms. Here, we also found the most common imaging features for the control group were flattening and sclerosis in initial stages, which does not correspond to OA findings; however, a significant number of patients presented cysts, erosions, or osteophytes (84%), leading to question whether the control patients truly presented with osteoarthrosis or the high-resolution of the images is what led to over-diagnosis when using the general guidelines from RDC/TMD.

This study also found the most common radiological features presented in both groups which were flattening and sclerosis. These results agree with the general literature and with the DC/TMD criteria, where those signs are not determinant for osteoarthritis/osteoarthrosis and are commonly seen due to adaptative responses and the aging process. Emshoff et al. [19] reported a rate of 21% of patients showing erosions without clinical symptoms and a study by dos Anjos et al. [30] showed that in a population treated for orthodontic purposes (n = 382) only 3% presented osteophytes, 0.5% erosions, and 0.8% cysts. In our study, we found approximately 48% with osteophytes, 25% erosions, and 17% cysts, using CBCT images with 1mm slice thickness. It is possible that this discrepancy is based on image resolution since both studies were conducted by experienced radiologists. The challenge now appears on how to propose adequate diagnostic criteria for images with higher resolutions (0.08 mm x 0.08 mm x 0.08 mm of voxel size), while trying to avoid a radiographic over-diagnosis.

Our statistical comparison suggests that erosion, osteophytes, and cysts are more robust to differentiate the groups compared to flattening, which has a significant p-value, but not as significant as the others (Fig 1C). The control group showed more than 70% of condyles having flattening, 48% showed osteophytes, 64% with sclerosis, and only 17% with cysts. For the c-DJD group, flattening was present in 82% of the condyles, osteophytes in 66%, sclerosis in72%, erosions in 46% and cysts in 46%. The DC/TMD criteria does not use flattening and localized sclerosis as determinants for DJD, and our results support this recommendation due to the higher number of these two findings in the control group (Fig 2).

Jiang et al. [5], in 2015 assessed CBCT images with a isotropic voxel size of 0.5 mm in asymptomatic patients according to the chewing-side preference. They did not evaluate the presence of erosions, osteophytes, or cysts; however, the results indicated the occurrence of morphological changes in the TMJ region, suggesting an adaptive process which occurs when patients have a chewing-side preference. In comparison to our results, approximately 78% of all condyles presented with some degree of flattening (Fig 2), and one of the reasons may also be due to those adaptative responses to abnormal functions.

As a limitation of this study, we did not perform a dosimetry study to determine and assess the X-ray absorption during our protocol for hr-CBCT acquisition. Lukat et al. [31], evaluated the effective dose of the CBCT for the TMJ region with a small FOV (5x3.7 cm); but, they used a different CBCT machine/protocol, and reported unilateral doses of 20.5 ± 1.3 μSv. Another limitation of this study is that we define as high-resolution the CBCT with a voxel size of 0.08mmx0.08mmx0.08mm; however, the spatial resolution is related to multiple factors, as described by Brulmann and Schulze [31]: ". . .The spatial resolution is related to the physical

pixel size of the sensor, the grey-level resolution, the reconstruction technique applied. . ." which we have not addressed.

## Conclusion

Our results suggest that additional imaging criteria for hr-CBCT may be needed to properly classify DJD since the results showed a high number of radiographic findings in the asymptomatic control group.

## Supporting information

**S1 File.**
(TXT)

## Author Contributions

**Conceptualization:** Jonas Bianchi, Antônio Carlos de Oliveira Ruellas, Lawrence M. Ashman, Erika Benavides, Fabiana Naomi Soki, Lucia Helena Soares Cevidanes.

**Data curation:** Jonas Bianchi.

**Funding acquisition:** Lucia Helena Soares Cevidanes.

**Investigation:** Jonas Bianchi, Lucia Helena Soares Cevidanes.

**Methodology:** Jonas Bianchi, Júlia Vieira Pastana Bianchi, Marilia Yatabe, Erika Benavides, Fabiana Naomi Soki, Lucia Helena Soares Cevidanes.

**Resources:** Lawrence M. Ashman.

**Supervision:** Lucia Helena Soares Cevidanes.

**Writing – original draft:** Jonas Bianchi, João Roberto Gonçalves, Antônio Carlos de Oliveira Ruellas, Marilia Yatabe, Lucia Helena Soares Cevidanes.

**Writing – review & editing:** Jonas Bianchi, João Roberto Gonçalves, Antônio Carlos de Oliveira Ruellas, Júlia Vieira Pastana Bianchi, Lawrence M. Ashman, Erika Benavides, Fabiana Naomi Soki, Lucia Helena Soares Cevidanes.

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
