## [Decision Letter · Decision Letter 0]

8 Apr 2021

PONE-D-21-00635

RADIOGRAPHIC INTERPRETATION USING HIGH-RESOLUTION CBCT TO DIAGNOSE DEGENERATIVE TEMPOROMANDIBULAR JOINT DISEASE

PLOS ONE

Dear Dr. Bianchi,

Thank you for submitting your manuscript to PLOS ONE. After careful consideration, we feel that it has merit but does not fully meet PLOS ONE’s publication criteria as it currently stands. Therefore, we invite you to submit a revised version of the manuscript that addresses the points raised during the review process.

We look forward to receiving your revised manuscript.

Kind regards,

Farhan Bazargani, DDS, PhD

Academic Editor

PLOS ONE

Reviewers' comments:

Reviewer's Responses to Questions

**Comments to the Author**

1. Is the manuscript technically sound, and do the data support the conclusions?

Reviewer #1: Partly

Reviewer #2: Partly

Reviewer #3: Partly

2. Has the statistical analysis been performed appropriately and rigorously? 

Reviewer #1: No

Reviewer #2: Yes

Reviewer #3: Yes

3. Have the authors made all data underlying the findings in their manuscript fully available?

Reviewer #1: Yes

Reviewer #2: Yes

Reviewer #3: Yes

4. Is the manuscript presented in an intelligible fashion and written in standard English?

Reviewer #1: Yes

Reviewer #2: Yes

Reviewer #3: No

5. Review Comments to the Author

Reviewer #1: Point 01

“However, the sensitivity and specificity for this diagnosis are only 0.55 and 0.61, respectively.”

What is(are) the reference(s) for this information?

Point 02

In the “sample size calculation” it is written that “We performed a post-hoc analysis for the comparison between groups and the variables using the software G-power”

How come that did the author perform a post-hoc analysis for sample size calculation if post-hoc analyses are conducted using data that has already been collected, and the sample size is supposed to be calculated before the experimental part of the study begins?

Point 03

Still for the “sample size calculation”:

“for the sum of the flattening values, 97% - osteophytes, 70% - sclerosis; 99% erosion and 99% for the cysts”

Where did these values come from? This was not explained.

Point 04

The following paragraph is repeated in the text:

“All participants were between 21 – 70 years old, with no history of systemic diseases, jaw joint trauma, surgery or recent jaw joint injections, current pregnancy, or congenital bone or cartilage disease. They were clinically evaluated by the same temporomandibular joint specialist, using the clinical signs and symptoms of the Diagnostic Criteria for Temporomandibular Disorders (DC/TMD). To be clinically diagnosed with degenerative joint disease (c-DJD group), they must present TMJ noise during movement or function in the last 30 days, and crepitus detected during mandibular excursive movements. In addition, they must report TMJ pain for less than ten years. On the other hand, for the control group, the subjects did not present any clinical signs and symptoms of DJD.”

Point 05

“The X-rays for both groups were taken”

Since when x-rays are “taken”?

Point 06

What is the meaning of “corrected” sagittal and coronal planes?

Point 07

The authors used Mann-Whitney test to compare sums of categorical variables?

And why are the results shown in figure 1C presented in ranks?

Together with my point 02 above, I can only conclude that this study was not correctly conducted from the statistical point of view. The authors of this manuscript need to address these issues by looking for help of a professional statistician. These statistical issues may question the validity of the results.

And the sub-item “Statistical analysis” from the Materials and Methods section is extremely poorly explained.

Point 08

In the Discussion, the paragraph beginning with “Interestingly, (…)” is a mere repetition of the results.

Point 09

The limitations of the study were neither listed nor discussed.

Point 10

“This study evaluated the radiographic findings based on the DC/TMD criteria for DJD in hr-CBCT images, and we”

Remove this text from the conclusion.

Reviewer #2: 1. The authors might wish to read and cite “Small field-of-view cone beam CT temporomandibular joint imaging dosimetry” (Dentomaxillofac. Radiol. 42: 20130082; 2013) and “The effect of voxel size on cone beam CT images of the temporomandibular joints” (Oral Surg., Oral Med., Oral Pathol., Oral Radiol. 119:229-237; 2015) from the Toronto group.

2. Please clarify the voxel dimensions of the Morita CBCT device. Is the product of length, width and height of the voxel equal to 0.08 mm^3, or is each individual dimension 0.08 mm (i.e., so the voxel size is 0.08 mm x 0.08 mm x 0.08 mm, and the volume is 0.000512 mm^3]. When I look at the Morita website, it is the latter (0.08 mm per side) and not what the authors have indicated. I believe this same error propagates into the Discussion as well when they cite the work of others.

3. How were the subjects recruited? Was this sequential enrollment of all patients who were seen in a TMD clinic between Jan 2016 and July 2019? Or were these a subset of the patients? If the latter, why were they chosen to be included. Please clarify.

4. Why did clinically asymptomatic control patients receive TMJ imaging? What was the justification for this?

5. Define “radiological experts”.

6. Table 1. Can the authors please explain what a “Confusion matrix” is and how the numbers were derived? Are there units for these numbers or are the units arbitrary? Please clarify.

7. Figure 1C. Please provide units of measurement along the y-axis of the graphs (even if they are Arbitrary Units; AU).

8. Is it necessary to report frequency to one-one-hundredths of a percentage?

9. A significant weakness of this study is that the authors did not perform any dosimetry for this imaging protocol, and inappropriately cited the work of others. There is considerable variation in calculated effective radiation doses for small field size CBCT fields, and the calculation of effective dose depends on the location of the field and the tissues contained in the volume. The Ludlow meta-analysis (reference 21) cites doses of 32 uSv in the maxilla and 43 uSv in the mandible. But Pauwels (who originally reported the 43 uSv mandible value) indicates that the field was centered in the “Lower jaw molar region” and not on the TMJ. So the dose from the current study where the image volume is centered on the TMJ is more than likely not 43 uSv. If you read the Toronto paper (in DMFR 2013), they reported unilateral doses of about 10 uSv and bilateral doses of about 21 uSv; much less than what Pauwels reported for the posterior mandible. Although the Toronto group used a 5 cm x 3.7 cm field size, their volumes were centered over the joints, and so should provide much more accurate values than Pauwels. However, without having done any dosimetry work themselves for this application of the 4 cm x 4 cm field, the authors can only speculate about the dose.

10. In the last paragraph of the Discussion, the authors bring in the topic of “radiomics”, but do not expand on the topic. I’m not sure how radiomics relates to the current work.

11. I am somewhat concerned about the reliability of the radiologic interpretation of normal and disease, although the consistency appears to be very good. In Figure 4D, I would say that this arrow is simply pointing to a large marrow space. There are other such spaces on the same image throughout the imaged portion of the cancellous bone, and some are perhaps even larger; this is not a subchondral cyst. In 4E, this may be an erosion (although the shape is not quite what I would expect for an erosion. More likely, this could be a small neurovascular canal. You can only truly tell by scrolling "in and out" of this cut. In 4F, I would argue that the “Osteophyte” in the control group is simply the edge of the mandibular fovea; the raised bony border where the lateral pterygoid muscle inserts into the bone. Some individuals have more prominent fovea than others which is why you might not see this in 100% of patients.

12. I think the conclusion over-reaches the scope of the findings. The results indicate very good consistency for the detection of osseous abnormalities, but that is all. Without having data from a larger voxel size system, it is not possibly to take the next step to suggest that “hr” CBCT is any better or worse than current CBCT protocols that use larger voxel sizes (again, see the Toronto publication from 2015 in 4O).

Reviewer #3: 1. Diagnostic Criteria for Temporomandibular Disorders (DC/TMD) (Schiffman et al.) considers various factors such as pain, headache, and disc displacment. In addition, it is presented for the purpose of simpler screening of clinical factors such as pain (not diagnosis).

However, in this paper, it is based only on pain and sound, and the criteria are not clear. This paper is only thesis that confirmed the bone changes by taking CBCT in the group with pain and sound and the group without sound.

2. CT slice thickness is not a spatial resolution.

3. If you have a 4x4 cm FOV, you could not take both TMJs at a time. Is the suggested effective dose doubled? It is difficult to accurately acquire the TMJ complex. Was it filmed without a sout image?

4.0.08mm3 is high resolution? The smaller the voxel size, the greater the noise. Why did you use 0.08mm3?

5. The image interpretation does not seem to have been evaluated correctly. In particular, when c of the control grouup is about to adaptation state, when the scelrosis of the trabecular pattern begins, the existing marrow space is emphasized, but not cyst.

6.What kind of opinion did you take when consent was not reached between the reviewers of the video?

6. PLOS authors have the option to publish the peer review history of their article (what does this mean?). If published, this will include your full peer review and any attached files.

Reviewer #1: No

Reviewer #2: No

Reviewer #3: No

---

## [Author Response · Author response to Decision Letter 0]

30 May 2021

Dear editor and reviewers. It is with great happiness that I’m submitting this revised version of our manuscript. The comments and observations were crucial to improve this study. We appreciate the time spent by the reviewers. We hope to have provided answers to your questions, and we did the modifications as requested in our revised manuscript.

Reviewer #1: Point 01

“However, the sensitivity and specificity for this diagnosis are only 0.55 and 0.61, respectively.” What is(are) the reference(s) for this information?

Thank you for this observation. We forgot to add this reference, and we appreciate your attention. We have updated the manuscript with the reference as follows:

14: Schiffman E, Ohrbach R, Truelove E, Look J, Anderson G, Goulet J-P, et al. Diagnostic Criteria for Temporomandibular Disorders (DC/TMD) for Clinical and Research Applications: recommendations of the International RDC/TMD Consortium Network* and Orofacial Pain Special Interest Group†. J oral facial pain headache. 2014;28: 6–27. doi:10.11607/jop.1151

In the above manuscript, the authors state on page 12: “… Validity of the Newly Recommended DC/TMD Axis I Diagnostic Algorithms Sufficient: The sensitivity of the recommended clinical criteria for DJD was 0.55 (0.47, 0.62) and specificity was 0.61 (0.56, 0.65)….”

Point 02

In the “sample size calculation” it is written that “We performed a post-hoc analysis for the comparison between groups and the variables using the software G-power”

How come that did the author perform a post-hoc analysis for sample size calculation if post-hoc analyses are conducted using data that has already been collected, and the sample size is supposed to be calculated before the experimental part of the study begins?

Thank you again for this valuable observation. We added the proper sample size calculation and excluded the post-hoc analysis after consulting a statistician.

Point 03

Still for the “sample size calculation”:

“for the sum of the flattening values, 97% - osteophytes, 70% - sclerosis; 99% erosion and 99% for the cysts” Where did these values come from? This was not explained.

We have fixed this statement after adding our sample size calculation.

Point 04

The following paragraph is repeated in the text:

“All participants were between 21 – 70 years old, with no history of systemic diseases, jaw joint trauma, surgery or recent jaw joint injections, current pregnancy, or congenital bone or cartilage disease. They were clinically evaluated by the same temporomandibular joint specialist, using the clinical signs and symptoms of the Diagnostic Criteria for Temporomandibular Disorders (DC/TMD). To be clinically diagnosed with degenerative joint disease (c-DJD group), they must present TMJ noise during movement or function in the last 30 days, and crepitus detected during mandibular excursive movements. In addition, they must report TMJ pain for less than ten years. On the other hand, for the control group, the subjects did not present any clinical signs and symptoms of DJD.”

Sorry for this typo. We have removed it from our corrected paper.

Point 05

“The X-rays for both groups were taken”

Since when x-rays are “taken”?

We have not noticed this informal language during our initial submission. The corrected manuscript is corrected as follows: “… The cone-beam computed tomography exams for both groups were acquired…” 

Point 06

What is the meaning of “corrected” sagittal and coronal planes?

The radiologist's experts performed a spatial orientation in the images when needed (sectional views) to best fit the TMJ region. We have removed the term “corrected” from our manuscript since it does not add valuable information. Thank you for noticing this. 

Point 07

The authors used Mann-Whitney test to compare sums of categorical variables?

And why are the results shown in figure 1C presented in ranks?

Together with my point 02 above, I can only conclude that this study was not correctly conducted from the statistical point of view. The authors of this manuscript need to address these issues by looking for help of a professional statistician. These statistical issues may question the validity of the results.

And the sub-item “Statistical analysis” from the Materials and Methods section is extremely poorly explained.

Thank you for this observation; we have consulted a statistician as well as requested. We are sorry that we did not express how we treated the data for the Mann-Whitney test. In this revised version, we hope that we could clarify your concerns and why the values are presented in Ranks. Also, we realized that our title for figure 1 was wrong. We have stated that we used the sum of the variables, which did not occur for the Mann-Whitney test (we used the actual values – quantitative ordinal variables). We have corrected the title of figure 1 as well. 

As an additional explanation, in our study, for the Mann-Whitney analysis specifically, we have used the variables as quantitative ordinal variables (0 to 3 on an ordinal scale) for each group. The Mann-Whitney results display the ‘Ranks' (that’s why the bar shows values from 0 to 200. Those values are “ranked values” given by the Mann-Whitney test and not the variables' values nor the sum of them. In summary, the logic behind the Mann-Whitney test is to rank the data for each condition and then see how different the two rank totals are. If there is a systematic difference between the two conditions, then most of the high ranks will belong to one condition, and most of the low ranks will belong to the other one. As a result, the rank totals will be quite different. On the other hand, if the two conditions are similar, then high and low ranks will be distributed fairly evenly between the two conditions, and the rank totals will be fairly similar.

Point 08

In the Discussion, the paragraph beginning with “Interestingly, (…)” is a mere repetition of the results.

You are right. We moved this paragraph to our results section because it summarizes important findings that are in figure 3 as well. Thank you for the observation and careful reading of our paper.

Point 09

The limitations of the study were neither listed nor discussed.

We have added limitations in our discussion as suggested by you and others reviewers. Thank you.

Point 10

“This study evaluated the radiographic findings based on the DC/TMD criteria for DJD in hr-CBCT images, and we”

Remove this text from the conclusion.

It was removed, thank you. 

Reviewer #2: 1. The authors might wish to read and cite “Small field-of-view cone beam CT temporomandibular joint imaging dosimetry” (Dentomaxillofac. Radiol. 42: 20130082; 2013) and “The effect of voxel size on cone beam CT images of the temporomandibular joints” (Oral Surg., Oral Med., Oral Pathol., Oral Radiol. 119:229-237; 2015) from the Toronto group.

Thank you. We read the papers and added them to our manuscript (discussion section). 

2. Please clarify the voxel dimensions of the Morita CBCT device. Is the product of length, width and height of the voxel equal to 0.08 mm^3, or is each individual dimension 0.08 mm (i.e., so the voxel size is 0.08 mm x 0.08 mm x 0.08 mm, and the volume is 0.000512 mm^3]. When I look at the Morita website, it is the latter (0.08 mm per side) and not what the authors have indicated. I believe this same error propagates into the Discussion as well when they cite the work of others.

You are right. The voxel size is 0.08 mm x 0.08 mm x 0.08 mm (each individual dimension). We have corrected through our text this important information. Thank you for your observation. 

3. How were the subjects recruited? Was this sequential enrollment of all patients who were seen in a TMD clinic between Jan 2016 and July 2019? Or were these a subset of the patients? If the latter, why were they chosen to be included. Please clarify.

This was a sequential enrollment of the patients with the TMD specialist. We have added this information to the text.

4. Why did clinically asymptomatic control patients receive TMJ imaging? What was the justification for this?

They were recruited as part of a large prospective study; the hr-CBCT was just one procedure; they also collected saliva, blood sample, and clinical parameters. This manuscript aims to assess only the hr-CBCT images (and clinical parameters for inclusion criteria). All participants have signed an informed consent term to participate in the research, and ethical approval was obtained from the University of Michigan.

5. Define “radiological experts.”

They are two dental Professors with a degree (residency/masters) in radiology (Radiologists). We added into the manuscript the information below, summarized in our methods section. Thank you for noticing this. We agree that this information is essential as well to the readers.

Expert 1: Clinical Associate Professor in the Department of Periodontics and Oral Medicine at the University of Michigan School of Dentistry and a Diplomate and Vice-President of the American Board of Oral and Maxillofacial Radiology (ABOMR)

Expert 2: Clinical Assistant Professor of Oral and Maxillofacial Radiology in the Department of Periodontics and Oral Medicine and Division of Oral Pathology/Medicine/Radiology. Ph.D. in Oral Health Sciences and residency/master’s degree in Oral and Maxillofacial Radiology - University of Connecticut

6. Table 1. Can the authors please explain what a “Confusion matrix” is and how the numbers were derived? Are there units for these numbers or are the units arbitrary? Please clarify.

The correct term is Agreement Matrix. Sorry for this mistake. We have corrected it in our new version. The number showed in the matrix is the number of condyles with the imaging finding and its category. We added this information in the Table Title. 

7. Figure 1C. Please provide units of measurement along the y-axis of the graphs (even if they are Arbitrary Units; AU).

Thank you again for this observation. We have added the information as Ranked Values. For the Mann-Whitney U test, the values (y-axis) are transformed into ranked values for comparison purposes (this is the statistical approach that this specific test used to treat ordinal data); for this reason, the y-axis represents Ranked values; without a unit of measurement.

8. Is it necessary to report frequency to one-one-hundredths of a percentage?

We tried to be consistent with our decimal values, but we agree that there is no clinical relevance for one-one-hundredths of a percentage. We kept this in the text since it does not modify the results, but we appreciate your concerns. 

9. A significant weakness of this study is that the authors did not perform any dosimetry for this imaging protocol, and inappropriately cited the work of others. There is considerable variation in calculated effective radiation doses for small field size CBCT fields, and the calculation of effective dose depends on the location of the field and the tissues contained in the volume. The Ludlow meta-analysis (reference 21) cites doses of 32 uSv in the maxilla and 43 uSv in the mandible. But Pauwels (who originally reported the 43 uSv mandible value) indicates that the field was centered in the “Lower jaw molar region” and not on the TMJ. So the dose from the current study where the image volume is centered on the TMJ is more than likely not 43 uSv. If you read the Toronto paper (in DMFR 2013), they reported unilateral doses of about 10 uSv and bilateral doses of about 21 uSv; much less than what Pauwels reported for the posterior mandible. Although the Toronto group used a 5 cm x 3.7 cm field size, their volumes were centered over the joints, and so should provide much more accurate values than Pauwels. However, without having done any dosimetry work themselves for this application of the 4 cm x 4 cm field, the authors can only speculate about the dose.

Thank you so very much for noticing this weakness and limitation of this study. Our goal was not to assess the radiation dose; however, we agree that our paper should define how we obtained this dosage and the limitations. We added this information/references in our discussion as limitation. We also add the information in our methodology so that the reader can see the information in the methods section. 

10. In the last paragraph of the Discussion, the authors bring in the topic of “radiomics”, but do not expand on the topic. I’m not sure how radiomics relates to the current work.

We removed this sentence from the manuscript since it was not our main goal here to discuss radiomics. Thank you for observing this. 

11. I am somewhat concerned about the reliability of the radiologic interpretation of normal and disease, although the consistency appears to be very good. In Figure 4D, I would say that this arrow is simply pointing to a large marrow space. There are other such spaces on the same image throughout the imaged portion of the cancellous bone, and some are perhaps even larger; this is not a subchondral cyst. In 4E, this may be an erosion (although the shape is not quite what I would expect for an erosion. More likely, this could be a small neurovascular canal. You can only truly tell by scrolling "in and out" of this cut. In 4F, I would argue that the “Osteophyte” in the control group is simply the edge of the mandibular fovea; the raised bony border where the lateral pterygoid muscle inserts into the bone. Some individuals have more prominent fovea than others which is why you might not see this in 100% of patients.

Thank you for this comment. We are sorry that we have chosen those “borderline” images. Your comments go towards the same comments that the radiologists' experts had when they were doing the radiological analysis. Unfortunately, Fig.4 is a statistic screenshot that has limitations for diagnosis purposes if only assessed by that single cross-sectional view; we agree that some findings may be related to anatomical variations, but at the same time, the radiologists have spent several hours scrolling the image up and down to make the final diagnosis, leading to the good agreement. We agree with your suggestions, and we changed figure 4 to show the imaging findings more clear.

 12. I think the conclusion over-reaches the scope of the findings. The results indicate very good consistency for the detection of osseous abnormalities, but that is all. Without having data from a larger voxel size system, it is not possibly to take the next step to suggest that “hr” CBCT is any better or worse than current CBCT protocols that use larger voxel sizes (again, see the Toronto publication from 2015 in 4O).

We agree. Our results do not support the previous conclusion. A comparative analysis with a large voxel size would be a future study for our group that may be helpful to make additional conclusions. We have adaptated our conclusion to be more concise with our results. Thank you.

Reviewer #3: 

1. Diagnostic Criteria for Temporomandibular Disorders (DC/TMD) (Schiffman et al.) considers various factors such as pain, headache, and disc displacment. In addition, it is presented for the purpose of simpler screening of clinical factors such as pain (not diagnosis).

However, in this paper, it is based only on pain and sound, and the criteria are not clear. This paper is only thesis that confirmed the bone changes by taking CBCT in the group with pain and sound and the group without sound.

Thank you so very much for your comment. Here, the patients were prior screening with pain, headache, TMJ sounds (such as disc displacement) to be part of the degenerative joint disease group, as we stated in the Clinical diagnosis, exclusion, and inclusion criteria section. In this paper, we aimed to show how a high-resolution CBCT can influence the imaging diagnosis, and that can be observed mainly based on the large number of imaging findings in the control group.

2. CT slice thickness is not a spatial resolution.

Sorry for this mistake. We have corrected this in the updated version of the paper. We have simplified the use of spatial resolution as slice thickness, but according to Brullmann and Schulze: “… The spatial resolution is related to the physical pixel size of the sensor, the grey-level resolution, the reconstruction technique applied, and some other factors…” We also added this information to our discussion.

3. If you have a 4x4 cm FOV, you could not take both TMJs at a time. Is the suggested effective dose doubled? It is difficult to accurately acquire the TMJ complex. Was it filmed without a sout image?

Yes, we have acquired one TMJ per time; we have added this missing information. We believe the CBCT technician has used one scout for positioning of the head. We added more information about the radiation dose in the discussion, limitations of the study. 

4.0.08mm3 is high resolution? The smaller the voxel size, the greater the noise. Why did you use 0.08mm3?

Thank you for this question. The noise here was reduced using a limited FOV and adequate CBCT sensor/parameters. Also, we used the term high-resolution because it is higher resolution than conventional CBCT (0.5 mm voxel size), but we have added in our discussion this topic. We understand that many factors affect spatial resolution, and we hope to provide more information to the reader in this revised manuscript version.

5. The image interpretation does not seem to have been evaluated correctly. In particular, when c of the control grouup is about to adaptation state, when the scelrosis of the trabecular pattern begins, the existing marrow space is emphasized, but not cyst.

Thank you for this comment. We are sorry that we have chosen those “borderline” images. Your comments go towards the same comments that the radiologists' experts had when they were doing the radiological analysis. Unfortunately, Fig.4 is a statistic screenshot that has limitations for diagnosis purposes if only assessed by that single cross-sectional view; we agree that some findings may be related to anatomical variations, but at the same time, the radiologists have spent several hours scrolling the image up and down to make the final diagnosis, leading to the good agreement. We agree with your suggestions, and we changed figure 4 to show the imaging findings more clear.

6.What kind of opinion did you take when consent was not reached between the reviewers of the video?

Thank you for these comments. We did not take an opinion from them; after the blinded and separated radiological classification, the radiologists re-evaluated all the images that they disagreed with and got a final consensus based on their expertise and agreement.

---

## [Decision Letter · Decision Letter 1]

2 Jul 2021

PONE-D-21-00635R1

RADIOGRAPHIC INTERPRETATION USING HIGH-RESOLUTION CBCT TO DIAGNOSE DEGENERATIVE TEMPOROMANDIBULAR JOINT DISEASE

PLOS ONE

Dear Dr. BIANCHI,

Thank you for submitting your manuscript to PLOS ONE. After careful consideration, we feel that it has merit but does not fully meet PLOS ONE’s publication criteria as it currently stands. Therefore, we invite you to submit a revised version of the manuscript that addresses the points raised during the review process.

We look forward to receiving your revised manuscript.

Kind regards,

Lucinda Shen

Staff Editor 

on behalf of 

Farhan Bazargani, DDS, PhD

Academic Editor

PLOS ONE

Journal Requirements:

Additional Editor Comments:

Dear Dr. Bianchi and co-workers,

It's my pleasure to inform you that your manuscript has been accepted for publication in the PLOS ONE.

Reviewers' comments:

Reviewer's Responses to Questions

**Comments to the Author**

1. If the authors have adequately addressed your comments raised in a previous round of review and you feel that this manuscript is now acceptable for publication, you may indicate that here to bypass the “Comments to the Author” section, enter your conflict of interest statement in the “Confidential to Editor” section, and submit your "Accept" recommendation.

Reviewer #1: All comments have been addressed

Reviewer #2: (No Response)

2. Is the manuscript technically sound, and do the data support the conclusions?

Reviewer #1: Yes

Reviewer #2: Partly

3. Has the statistical analysis been performed appropriately and rigorously? 

Reviewer #1: Yes

Reviewer #2: Yes

4. Have the authors made all data underlying the findings in their manuscript fully available?

Reviewer #1: Yes

Reviewer #2: Yes

5. Is the manuscript presented in an intelligible fashion and written in standard English?

Reviewer #1: Yes

Reviewer #2: No

6. Review Comments to the Author

Reviewer #1: The manuscript now seems to be suitable for publication. The authors were able to address all the issues raised.

Reviewer #2: Comments from the original review:

1. Comment 5. The word “expert” is a difficult one to quantify and so should be avoided. When referring to the individuals who viewed the CBCT images, I would suggest you the term “board-certified oral and maxillofacial radiologist”.

2. Comment 9. I would remove any and all references to dosimetry since i) it was never done; and ii) you cannot apply the dosimetry of one system to another, particularly if the center of the imaging volume is not the same. I believe I made mention of this in my original comments.

New comments:

1. The use of English in this revision reads more poorly than the initial submission. It requires some work.

2. Abstract. ‘k’ is Cohen’s kappa? If ‘yes’, then “kappa” should be spelled out (with the name Cohen) or the appropriate Greek letter should be used.

3. If you tell us the voxel is 0.08 mm, you should add a word or phrase to indicate that the voxel is isotopic since you only give 1 dimension of the 3.

4. There should be alignment between the Objective/Aim and what is being concluded. This does not occur in the Abstract.

5. Introduction. CBCT imaging is not a “dental research” tool, so the first sentence in the second paragraph is a strange assertion.

6. In the last paragraph of the Introduction, an Aim or Objective, or better yet, a hypothesis should be included. Providing insight” is not scientific.

7. Materials and Methods. What is “STROBE”?

8. Again, reference is incorrectly made to 43 uSv as the dose for each TMJ in the Materials and Methods section in the paragraph “High-resolution CBCT radiation information”. For the reasons I gave in the original review about dosimetric calculations, 43 uSv is misleading and erroneous. I don't know why such references continue in this revised version.

9. To be clear, this is the Department of [MEDICAL] Radiology at the University of Michigan?

10. The references numbers do not align with the correct references in the text.

11. I am still suspicious of the validity of the radiologic observations. In Figure 4A, the “Cyst” is not round or hydraulic or smooth (as cysts should be). This is not a cyst but likely a marrow space like the other similarly appearing marrow spaces in the condylar neck.

12. In Figure 4B, how can an erosion be located along the anterior surface of the condylar head since there is no biomechanical loading in this area.

13. Figure 4E is a nice example of a subchondral cyst; it is not an erosion because I can resolve a cortex that is intact.

7. PLOS authors have the option to publish the peer review history of their article (what does this mean?). If published, this will include your full peer review and any attached files.

Reviewer #1: No

Reviewer #2: No

---

## [Author Response · Author response to Decision Letter 1]

23 Jul 2021

We thank you to all the three reviewers and editors that spent time and effort helping to improve this manuscript. We certainly have increased the quality of our data after your suggestions and comments. 

Sincerely,

Authors.

Reviewer #2: 

We thank you so much for your comments and observations. They improved the quality of our work. We greatly appreciate your time and careful analysis of our study, and we hope to provide the answers to your questions/suggestions.

1. Comment 5. The word "expert" is a difficult one to quantify and so should be avoided. When referring to the individuals who viewed the CBCT images, I would suggest you the term "board-certified oral and maxillofacial radiologist".

We have addressed the changes as requested. We are sorry that we have not corrected this properly in the previous review. 

2. Comment 9. I would remove any and all references to dosimetry since i) it was never done; and ii) you cannot apply the dosimetry of one system to another, particularly if the center of the imaging volume is not the same. I believe I made mention of this in my original comments.

We agree with the reviewer, we have added as a limitation previously, but we agree that using this in our methodology can lead the readers to the wrong perception. Therefore, as we have not performed a dosimetry study, we have removed the information from our manuscript, as suggested. 

1. The use of English in this revision reads more poorly than the initial submission. It requires some work.

We did a general English correction in the entire paper with our department's native English speaker staff; the changes are tracked. 

2. Abstract. 'k' is Cohen's kappa? If 'yes', then "kappa" should be spelled out (with the name Cohen) or the appropriate Greek letter should be used.

Thank you for this critical observation. We have corrected this not only in the abstract but in the entire manuscript as well. 

3. If you tell us the voxel is 0.08 mm, you should add a word or phrase to indicate that the voxel is isotopic since you only give 1 dimension of the 3.

Thank you. In this revised version, we added this information. 

4. There should be alignment between the Objective/Aim and what is being concluded. This does not occur in the Abstract.

We wrote the conclusion again, and we hope to be now better aligned with our aim. 

5. Introduction. CBCT imaging is not a "dental research" tool, so the first sentence in the second paragraph is a strange assertion.

We changed this paragraph as follows: "In dentistry, CBCT is the exam of choice to evaluate many bone-related diseases."

6. In the last paragraph of the Introduction, an Aim or Objective, or better yet, a hypothesis should be included. Providing insight" is not scientific.

We agree; we have now stated our hypothesis. 

7. Materials and Methods. What is "STROBE"?

We have added this information in the material and methods section. (Strengthening the Reporting of Observational Studies in Epidemiology (STROBE)) 

8. Again, reference is incorrectly made to 43 uSv as the dose for each TMJ in the Materials and Methods section in the paragraph "High-resolution CBCT radiation information". For the reasons I gave in the original review about dosimetric calculations, 43 uSv is misleading and erroneous. I don't know why such references continue in this revised version.

Thank you again for this observation; we now have removed this information since this was not our aim, and we did not perform a dosimetry study. However, we also have included the manuscript provided by you from the Toronto group in our discussion as additional information. 

9. To be clear, this is the Department of [MEDICAL] Radiology at the University of Michigan?

This was the Department of Periodontics & Oral Medicine at the University of Michigan – School of Dentistry. We have corrected this in our manuscript. Thank you for this observation; we have not noticed this typo before because both professors (radiologists) teach radiology at the school.

10. The references numbers do not align with the correct references in the text.

We did an overall review of the references/numbers. The problems occurred during the track version changes / clean version and reference manager. 

11. I am still suspicious of the validity of the radiologic observations. In Figure 4A, the "Cyst" is not round or hydraulic or smooth (as cysts should be). This is not a cyst but likely a marrow space like the other similarly appearing marrow spaces in the condylar neck.

Thank you for your observation; you are right. I have consulted the two radiologists who agreed with your comments. For this reason, I have chosen another case to illustrate the radiologist's findings.

12. In Figure 4B, how can an erosion be located along the anterior surface of the condylar head since there is no biomechanical loading in this area.

I'm sorry, I have added the arrow in the wrong direction. As I do not have the original PowerPoint to modify the arrow position, I choose another image to replace 4B. Thank you.

13. Figure 4E is a nice example of a subchondral cyst; it is not an erosion because I can resolve a cortex that is intact.

After consulting the radiologists, they also agree with you. For this reason, I have chosen another case to illustrate the erosion in the control group.

---

## [Editor Report · Decision Letter 2]

28 Jul 2021

RADIOGRAPHIC INTERPRETATION USING HIGH-RESOLUTION CBCT TO DIAGNOSE DEGENERATIVE TEMPOROMANDIBULAR JOINT DISEASE

PONE-D-21-00635R2

Dear Dr. Bianchi,

We’re pleased to inform you that your manuscript has been judged scientifically suitable for publication and will be formally accepted for publication once it meets all outstanding technical requirements.

Kind regards,

Farhan Bazargani, DDS, PhD

Academic Editor

PLOS ONE

---

## [Editor Report · Acceptance letter]

2 Aug 2021

PONE-D-21-00635R2 

Radiographic Interpretation Using High-Resolution Cbct to Diagnose Degenerative Temporomandibular Joint Disease. 

Dear Dr. Bianchi:

I'm pleased to inform you that your manuscript has been deemed suitable for publication in PLOS ONE. Congratulations! Your manuscript is now with our production department. 

Kind regards, 

on behalf of

Dr. Farhan Bazargani 

Academic Editor

PLOS ONE